# Are There Any Changes in the Causative Microorganisms Isolated in the Last Years from Hip and Knee Periprosthetic Joint Infections? Antimicrobial Susceptibility Test Results Analysis

**DOI:** 10.3390/microorganisms11010116

**Published:** 2023-01-01

**Authors:** Mihai Dan Roman, Bogdan-Axente Bocea, Nicolas-Ionut-Catalin Ion, Andreea Elena Vorovenci, Dan Dragomirescu, Rares-Mircea Birlutiu, Victoria Birlutiu, Sorin Radu Fleaca

**Affiliations:** 1Faculty of Medicine Sibiu, Lucian Blaga University, Str. Lucian Blaga, Nr. 2A, 550169 Sibiu, Romania; 2County Clinical Emergency Hospital, 550245 Sibiu, Romania; 3Economic Cybernetics and Statistics Doctoral School, Bucharest University of Economic Studies, Piata Romana 6, 010371 Bucharest, Romania; 4Clinical Hospital of Orthopedics, Traumatology, and Osteoarticular TB, B-dul Ferdinand 35–37, Sector 2, 021382 Bucharest, Romania

**Keywords:** periprosthetic joint infection, infection, microbiology, pathogen, etiology, trend, antimicrobial susceptibility test

## Abstract

Background: PJIs following total hip and knee arthroplasty represent severe complications with broad implications, and with significant disability, morbidity, and mortality. To be able to provide correct and effective management of these cases, an accurate diagnosis is needed. Classically, acute PJIs are characterized by a preponderance of virulent microorganisms, and chronic PJIs are characterized by a preponderance of less-virulent pathogens like coagulase-negative staphylococci or *Cutibacterium* species. This paper aims to analyze if there are any changes in the causative microorganisms isolated in the last years, as well as to provide a subanalysis of the types of PJIs. Methods: In this single-center study, we prospectively included all retrospectively consecutive collected data from patients aged over 18 years that were hospitalized from 2016 through 2022, and patients that underwent a joint arthroplasty revision surgery. A standardized diagnostic protocol was used in all cases, and the 2021 EBJIS definition criteria for PJIs was used. Results: 114 patients were included in our analysis; of them, 67 were diagnosed with PJIs, 12 were acute/acute hematogenous, and 55 were chronic PJIs. 49 strains of gram-positive aerobic or microaerophilic cocci and 35 gram-negative aerobic bacilli were isolated. Overall, *Staphylococcus aureus* was the most common isolated pathogen, followed by coagulase-negative staphylococci (CoNS). All cases of acute/acute hematogenous PJIs were caused by gram-positive aerobic or microaerophilic cocci pathogens. Both *Staphylococcus epidermidis* and methicillin-resistant *S. aureus* were involved in 91.66% of the acute/acute hematogenous PJIs cases. 21.8% of the chronic PJIs cases were caused by pathogens belonging to the Enterobacterales group of bacteria, followed by the gram-negative nonfermenting bacilli group of bacteria, which were involved in 18.4% of the cases. 12 chronic cases were polymicrobial. Conclusion: Based on our findings, empiric broad-spectrum antibiotic therapy in acute PJIs could be focused on the bacteria belonging to the gram-positive aerobic or microaerophilic cocci, but the results should be analyzed carefully, and the local resistance of the pathogens should be taken into consideration.

## 1. Introduction

There is a broad range of biofilm-related infections (BRI), from catheter-associated urinary tract infections (which still represent the most common BRIs) to central line-associated bloodstream infections, fracture-related infections, BRI associated with the use of fixed braces, and last but not least, periprosthetic joint infections [1,2,3]. 

Bacteria can be organized into two types: planktonic bacteria when they appear as single cell structures, and biofilms when they are structured in a sessile form and organized in multicellular aggregates [4,5]. To this day, we still do not have a consensus definition of a biofilm; but often, biofilms are defined as ‘a coherent cluster of bacterial cells imbedded in a biopolymer matrix, which, compared with planktonic cells, shows increased tolerance to antimicrobials and resists the antimicrobial properties of the host defence’ [5]. Bacteria are capable of a rapid transition between planktonic forms and biofilms [6,7,8]. One of the most important differences between planktonic bacteria and biofilm bacteria is represented by the antibiotic tolerance. Biofilm bacteria is known to present increased survival capabilities on exposure to multiple classes of antibiofilm antiobiotics. Characteristics that are related to many bacterial functions are the slow growth state, impaired expression of antimicrobial resistance mechanisms, and possible depletion aggregation [9,10,11,12,13]. Biofilms-related infections are recalcitrant to antibiotic strategies. It has been already published in the literature that the antimicrobial concentrations needed to eradicate biofilms are higher than the concentrations required to eradicate the same bacterial clones in a planktonica state [14].

Biofilms are commonly associated with a foreign body material, such as when a prosthetic is implanted; abiotic artificial surfaces that offer a perfect interface to which planktonic bacteria may attach and form a biofilm [15].

*Staphylococcus aureus* represents a frequent intra- and extracellular pathogen associated with orthopedic devices biofilm-related infections. In several publications in the literature, it is reported that biofilm-forming *S. aureus* is the most common pathogen in biofilm-related infections, as well as the main pathogen associated with reinfection, due to its high resistance to the immune response and antibiotic treatments, as well as to its ability to infect not only bone-forming cells (osteoblasts), but also the cells responsible for bone resorption (osteoclasts) [16]. The incidence of orthopedic methicillin-resistant *Staphylococcus aureus* infections has increased, and since it has also been proven effective against MRSA strains, vancomycin is recommended as the first-line antibiotic therapy choice for treatment of orthopedic MRSA infections [17].

Total joint arthroplasties are a very effective medical intervention. Unfortunately, complications may occur in some patients [18]. Periprosthetic joint infections (PJIs) are devastating complications following total joint arthroplasty, most commonly associated with total hip or knee arthroplasty (due to the increased number of this type of surgery), with broad implications, and with significant morbidity and mortality [18]. A total of 16.8% of all knee-revision surgeries and 14.8% of all hip-revision surgeries are due to failure caused by PJIs [19]. Prosthetic joint infections occur at a frequency of 1 to 3% and are still a major cause of healthcare expenditure [20]. Other authors report that periprosthetic joint infections of the hip and knee occur in approximately 1 to 2% of patients after total joint arthroplasties [21,22]. These are complications that lead to a prolonged hospital stay, multiple surgeries, and functional impairment [23]. Knowledge of the microbiological agent that causes the PJIs is one of the most important aspects, together with the antimicrobial susceptibility test (AST) results. Information that is also essential for guiding empiric antibiotic therapy, particularly in acute periprosthetic joint infections. Unfortunately, local data is not available. The aim of this paper is to analyze if there are any changes in the causative microorganisms isolated in the last years, as well as to provide a sub-analysis of the types of PJIs. We also analyzed the antimicrobial susceptibility test results especially to see if any changes in the frequency of antimicrobial-resistant organisms in PJIs have occurred in the last years.

## 2. Materials and Methods

### 2.1. Study Design

A single-center observational, cohort, ongoing study was conducted in the Emergency Clinical County Hospital, Romania. Before patients’ inclusion in the study, the study protocol was reviewed and approved by the institutional review board. A standardized diagnostic system was used to assess all patients who underwent surgical intervention for the revision of a joint prosthesis to determine implant failure. Our implemented diagnostic strategy included a sampling of intraoperative tissue specimens, sonication of the retreated implant and sonication fluid cultures, and cell counting of the synovial fluid. As a rapid method of bacteria detection from the sonication fluid, we used a bbFISH kit (hemoFISH Masterpanel, Miacom diagnostics GmbH Düsseldorf, Germany). All specimens were inoculated on aerobic and anaerobic culture media (Schaedler anaerobe broth, Sabouraud plate, MacConkey agar plate, glucose broth, lactose broth, and thioglycollate broth), and a 14-day period of incubation period was implemented as a standard.

### 2.2. Study Population

We prospectively included all consecutive patients, aged over 18 years, who were hospitalized from 2016 through October 2022 and underwent joint arthroplasty revision surgery for any reason. We excluded all cases with positive bacterial cultures from harvested specimens during the second stage of 2-stage revision surgery and all with positive sonication fluid cultures from spacer sonication. Detailed information was extracted from the medical records of the patients using a standardized electronic collection form. All data were available for all the enrolled patients.

### 2.3. Laboratory Studies

Our newly implemented diagnostic strategy included a standardized sampling of at least four intraoperative tissue samples (one of the samples used for the histopathological examination (the periprosthetic membrane) and the others were sent to the microbiological laboratory for bacterial cultures). For the sonication of the retrieved implants, in the operating theater, sterile Ringer’s or saline solution was added over the implants that were deposited in sterile containers. These containers were previously sterilized according to the manufacturer’s instructions and double-packed. The implants were processed within 30 min by sonication (1 min) using an ultrasound bath (BactoSonic14.2, Bandelin GmbH, Berlin, Germany) at a frequency of 42 kHz and a power density of 0.22 W/cm^2^. The resulting sonication fluid was vortexed, and 50 mL of sonication fluid was centrifuged at 2500 rpm for 5 min. The resulting precipitate was inoculated. If >50 CFU/mL were counted, sonication fluid cultures were considered positive. Ten milliliters of sonication fluid were incubated in blood culture bottles in a blood culture system (BD BACTEC™). Regarding the periprosthetic tissue cultures, tissue samples were collected in sterile vials and individually homogenized in 1 mL thioglycolate broth. Tissue homogenate samples (1 mL) were inoculated into the culture media. Synovial fluid was aspirated preoperatively in a native vial and inoculated into different media for culturing. All biological samples that required cultures were inoculated and incubated aerobically, anaerobically, and in a high concentration of CO^2^ (GENbag-GENbox Atmospheric generators bioMérieux, Marcy-l’Étoile, France) Schaedler anaerobe broth, Sabouraud plate, MacConkey agar plate, glucose broth, lactose broth, and thioglycollate broth, at 37 °C. The isolated bacteria were identified using a VITEK 2 Compact analyzer (bioMérieux, Marcy-l’Étoile, France). Minimum inhibitory concentrations were assessed according to the EUCAST (European Committee on Antimicrobial Susceptibility Testing) breakpoints. We were able to analyze cultures during working days and weekends. We previously published full details of the implemented protocol [24,25].

### 2.4. Study Definitions and Classification

A culture was marked as positive on the day that an isolate was identified by the VITEK 2 Compact analyzer (bioMérieux, Marcy-l’Étoile, France), the first day of growth. A periprosthetic joint infection was diagnosed using the 2021 European Bone and Joint Infection Society (EBJIS) definition for the diagnosis of periprosthetic joint infection (Table 1.) [26].

We used the classification proposed by Zimmerli et al. to determine if there was an acute, late chronic, or acute late periprosthetic joint infection, a classification that defines the prosthetic joint infections as early (occurring within 3 months after surgery), delayed (3–24 months), or late (>24 months) [20]. Due to the small number of enrolled patients, we also used a much simpler classification, a classification from the Pocket Guide to Diagnosis & Treatment of Periprosthetic Joint Infection (PJI) of the PRO-IMPLANT Foundation, Berlin, Germany (coordinated by N. Renz and A. Trampuz), a guide that is in line with national and international recommendations and that defines periprosthetic infections as acute or chronic (Perioperative/Hematogenous or per continuitatem).

### 2.5. Statistical Analysis

We performed the statistical analysis using the IBM SPSS Statistics^®^ version 28 software. Continuous variables were summarized as medians and interquartile ranges or mean and standard deviation and categorical variables as percentages of the total sample for that variable. Overall Percentages of culture positive PJIs were determined and estimated with a 95% CI. The Mantel-Haenszel chi-square test was used to determine if there was a statistically significant conditional dependence between the percentages of the identified PJI organisms and multidrug resistant bacteria over the studied period. A significance level of *p* ≤ 0.05 was used for all statistical tests. 

## 3. Results

A total number of 114 patients underwent debridement, antibiotics, and implant retention procedures or one-stage or two-stage revision surgery from 2016 through 2022. A diagnosis of aseptic loosening of an endoprosthesis was established in 40 adult patients during the study period. A total number of 67 episodes of periprosthetic joint infections were diagnosed in the analyzed period from 67 cases, cases that were culture-positive ones. We excluded seven cultures from the final analysis that were considered contaminants and six cases with positive cultures from harvested specimens during the second stage of 2-stage revision surgery. Three patients had culture-negative PJIs. A total of 67 confirmed PJIs were included in our final analysis. Of the 67 cases analyzed in our study, 12 had acute/acute hematogenous PJIs and 55 had chronic PJIs. Eight cases were acute PJIs and four were acute hematogenous PJIs. Due to the small number of enrolled patients, we decided to analyze the acute and acute hematogenous PJIs together. The 67 cases of PJIs included 40 hip prosthesis and 27 knee prosthesis. We were able to isolate 12 microorganisms from the 12 acute PJI cultures. The 55 chronic PJI cultures yielded 72 isolated microorganisms. We were able to group the 67 patients diagnosed with a periprosthetic joint infection, using the classification proposed by Zimmerli et al., as follows: ten patients were diagnosed with early PJI, nine patients with delayed PJI, and forty-eight patients were diagnosed with a late PJI. Again, using the classification of the periprosthetic joint infections proposed in the Pocket Guide by the PRO-IMPLANT Foundation as we did in previously published articles [22], eight patients were diagnosed with an acute perioperative infection, four patients with acute hematogenous infection, and fifty-five patients with chronic PJI. We will report all our results using this classification. Figure 1 represents the flow diagram showing details of the enrolled patients.

Characteristics of the enrolled patients in the study are outlined in Table 2. The mean age of the study population was 68.5 years old (±10.88 SD) and 34 (50.74%) were male patients. The median ASA score of the studied population was 2 and the median Charlson Comorbidity Index was 3. 92.53% of the enrolled patients had at least one comorbidity, the most common ones being arterial hypertension, heart failure, diabetes mellitus, chronic heart disease, peripheral vascular disease, and obesity. Osteoarthritis was the most common reason for primary prosthesis implantation (55.22%), followed by femoral neck fracture (10.44%), avascular necrosis (5.96%), and rheumatoid arthritis (2.98%). No trend changes in the baseline characteristics of the enrolled patients diagnosed with PJIs were found during the study period.

According to our results, 48 (55.2%; 95% CI 44.8–65.5) episodes of PJIs were caused by gram-positive aerobic or microaerophilic cocci and 35 (40.2%; 95% CI 29.9–50.6) by gram-negative aerobic bacilli. Overall, *Staphylococcus aureus* was the most common isolated pathogen, 21 (24.1%; 95% CI 14.9–33.3); followed by coagulase-negative staphylococci (CoNS), 19 (21.8%; 95% CI 13.8–31); Enterobacterales, 19 (21.8%; 95% CI13.8–31); and gram-negative nonfermenting bacilli, 16 (18.4%; 95% CI 10.3–26.4). 

A microbiological diagnosis was obtained in 67 cases: 23 cases in 2016–2017, 33 in 2018–2019, and 11 in 2020–2022. A significant variation in the proportion of cases with a microbiological diagnosis using our diagnostic method was not observed during the study period. A total of 12 cases of PJIs were polymicrobial and all of them were chronic cases. Additionally, no significant trends over the study period in the proportion of polymicrobial PJIs were observed.

Table 3 represents the lists of causative microorganisms of PJIs during the study period (2016 through 2021). Gram-positive aerobic or microaerophilic cocci were the most common group of isolated organisms, followed by gram-negative aerobic bacilli. A biennial proportion analysis of the isolated microorganisms was performed. Unfortunately, no changes during the study period were observed, and no statistically significant rising or decreasing linear trends were observed for PJIs caused by gram-positive aerobic or microaerophilic cocci or by gram-negative aerobic bacilli (Figure 2, Figure 3 and Figure 4).

All cases of acute/acute hematogenous PJIs were caused by gram-positive aerobic or microaerophilic cocci pathogens. Both *Staphylococcus epidermidis* and methicillin-resistant *Staphylococcus aureus* were involved in 91.66% of the acute/acute hematogenous PJIs cases. A total of 19 strains of pathogens from chronic PJIs cases belonged to the Enterobacterales group of bacteria, followed by the gram-negative nonfermenting bacilli group of bacteria, which were isolated in 16 samples.

48 strains of the isolated bacteria were multidrug-resistant bacterial strains (following the specified definition). Again, no statistically significant rising or decreasing linear trends were observed for multidrug-resistant bacteria.

### 3.1. Antimicrobial Susceptibility Test (AST) Results

We are reporting the AST based on the MICs (minimum inhibitory concentrations) of isolated bacterial strains that were evaluated following the European Committee on Antimicrobial Susceptibility Testing breakpoints (EUCAST) available at the time the strains were isolated. We will report the AST for the most isolated strains from our study. The AST results based on the MIC values are reported as follows: susceptible (S), intermediate (I), and resistant (R).

From the **Coagulase-negative *staphylococci*** (CoNS) species, eleven strains of *Staphylococcus epidermidis* were isolated, two strains of *Staphylococcus lentus*, three strains of *Staphylococcus xylosus*, one strain of *Staphylococcus hominis*, and two strains of *Staphylococcus haemolyticus*. Based on the MIC to oxacillin (4 mg/L–resistance), we can conclude that nine isolated strains were methicillin-resistant *Staphylococcus epidermidis* strains and two methicillin-susceptible *Staphylococcus epidermidis* strains. All strains of *Staphylococcus lentus*, *Staphylococcus hominis*, and *Staphylococcus haemolyticus* were also methicillin-resistant strains. All strains of *Staphylococcus xylosus* were methicillin-susceptible ones. A total of four CoNS strains were resistant to gentamycin and to quinolones/fluoroquinolones. All strains were susceptible to linezolid, teicoplanin, and vancomycin. All strains were also susceptible to trimethoprim/sulfamethoxazole, and just nine to rifampicin. According to EUCAST, it is known that vancomycin MIC values of 2 mg/L are on the border of the wild-type distribution and there may be an impaired clinical response. A MIC >2 mg/L for vancomycin measured using VITEK was not encountered, indicating that strains with reduced susceptibility to vancomycin were not isolated. We did not find an association between a MIC > 1 mg/L and co-resistance with rifampin: all rifampin-resistant strains (n = 2) had a vancomycin MIC <2 mg/L. Details regarding the AST results are reported in Table 4.

Of the 21 strains of *Staphylococcus aureus* that were analyzed in this study based on the MIC to oxacillin, 14 strains were methicillin-resistant strains. All strains of *Staphylococcus aureus* maintained their susceptibility to fluoroquinolones, teicoplanin, linezolid, trimethoprim/sulfamethoxazole, and to rifampicin. A MIC >1 mg/L for teicoplanin measured using VITEK was not encountered. We did not find an association between a MIC >1 mg/L and co-resistance with rifampin, all isolated strains were susceptible to rifampin. Details regarding the AST results are reported in Table 5.

Six strains of *Enterococcus faecalis* were isolated in our study, of them, four strains were high-level resistant to gentamicin and other aminoglycosides except for streptomycin, and three strains were high-level resistant to streptomycin. All strains were susceptible to linezolid, teicoplanin, vancomycin, and tigecycline. A total of four out of the six *Enterococcus faecalis*-isolated strains were susceptible to trimethoprim/sulfamethoxazole. Details regarding the AST results are reported in Table 6.

Six strains of *Escherichia coli* were isolated in our study and AST was performed in all cases. Susceptibility was analyzed to beta-lactams-dibactams and in this case to ureidopenicillins such as piperacillin/ticarcillin in combination with a beta-lactamase inhibitor (piperacillin/tazobactam; ticarcillin/clavulanic acid)—the sensitivity being preserved for all strains. In five out of the six strains, resistance to gentamicin was recorded. As for susceptibility to carbapenems, it was preserved in all three tested antibiotics (ertapenem, imipenem, and meropenem). From the group of cephalosporins, the use of those of the 3rd or 4th generation is at least questionable, at least from the point of view of susceptibility, with four strains of *Escherichia coli* isolated being resistant to cefotaxime, ceftazidime, and cefepime. Thus, two of the most frequently used classes of antibiotics in the management of infections associated with orthopedic implants remain under discussion; quinolones/fluoroquinolones and sulfamides, five strains being sensitive to ciprofloxacin, five to trimethoprim/sulfamethoxazole, and all strains being intermediately sensitive to norfloxacin. Susceptibility to fosfomycin is preserved. Details regarding the AST results are reported in Table 7.

From the *Enterobacter* species isolated strains, seven were identified as being *Enterobacter cloacae* complex and one was *Enterobacter amnigenus 2*. A total of five of the isolated strains were susceptible to Piperacillin/tazobactam, and four (50%) of the isolated strains were resistant to cephalosporins. 75% of the strains were resistant to gentamicin, three strains were resistant to ciprofloxacin, and susceptibility to fosfomycin was preserved for two strains. All strains maintained their susceptibility to trimethoprim/sulfamethoxazole. Details regarding the AST results are reported in Table 8.

Of the two strains of *Klebsiella* spp. isolated, one strain was susceptible to most of the antibiotics used to treat bone and joint infections, and the other one was just intermediate to tigecycline.

Details regarding the AST results of *Proteus mirabilis* are reported in Figure 5.

From the *Pseudomonas* species isolated strains, eight were identified as being *Pseudomonas aeruginosa* and one as *Pseudomonas fluorescens*. A total of five of the isolated strains were susceptible to ticarcillin/clavulanic acid, four to piperacillin/tazobactam, five to ceftazidime, six to imipenem, seven to meropenem, seven to ceftazidime/avibactam and ceftolozane/tazobactam, and five to ciprofloxacin. Details regarding the AST results are reported in Table 9.

A total of three strains of *Acinetobacter* species were identified in our study. Of them, all strains maintained their susceptibility to colistin and meropenem, and two strains maintained both to levofloxacin and trimethoprim/sulfamethoxazole. Details regarding the AST results are reported in Table 10.

A total of four strains of *Ralstonia picketti* species were identified in our study. Of them, all strains maintained their susceptibility to piperacillin, imipenem, meropenem, ciprofloxacin, pefloxacin, minocycline, and trimethoprim/sulfamethoxazole. All strains were resistant to tobramycin, and colistin. Details regarding the AST results are reported in Table 11.

### 3.2. Multidrug-Resistant Periprosthetic Joint Infections

A total of 48 isolated strains were multidrug-resistant bacteria strains (following the specified definition) during the study period, including 14 methicillin-resistant *S. aureus* (MRSA) and 17 multidrug-resistant strains of gram-negative aerobic bacilli. MRSA and multidrug-resistant gram-negative aerobic bacilli were simultaneously involved in five cases of PJIs. As previously mentioned, no statistically significant rising or decreasing linear trend was observed for multidrug-resistant bacteria. The following species accounted for 100% of all isolated strains of multidrug-resistant gram-negative aerobic bacilli: *Pseudomonas* spp., four strains; *Escherichia coli,* five strains; *Acinetobacter* spp., two strains; *Enterobacter cloacae* complex, three strains; and *Ralstonia pickettii*, three strains. A total of 10 of the 17 multidrug-resistant gram-negative aerobic bacilli strains that were isolated in our study were also extended-spectrum β-lactamases-producing Enterobacterales strains. Concerning the resistance to specific antibiotics used in the management of PJIs, the most relevant is the resistance of some species at ciprofloxacin among gram-negative aerobic bacilli (11 strains).

## 4. Discussion

This is the first study from Romania that reported on causative microorganisms isolated in the last years from hip and knee periprosthetic joint infections. Most cases of periprosthetic joint infections in our study are monomicrobial (55 cases), and *Staphylococcus aureus* is also the most common cause of infection in this study, with similar data being previously reported in the literature [27,28,29,30,31]. No statistically significant rising or decreasing linear trends were observed for PJIs caused by gram-positive aerobic or microaerophilic cocci or by gram-negative aerobic bacilli, and also no statistically significant rising or decreasing linear trends were observed for multidrug-resistant bacteria, which is surprising compared to the data published by Benitio N. et al. in 2016, in which the authors report an increase of the proportion of PJIs caused by aerobic gram-negative aerobic bacilli [31]. Additionally, the same group of authors report an increase of the number of cases of PJIs caused by fungi; in our study, no cases of fungal PJIs were diagnosed. Our results should be analyzed according also to the limitations of our study. A total of 48 isolated strains were multidrug-resistant bacteria strains during the study period, including 14 methicillin-resistant *S. aureus* (MRSA) and 17 multidrug-resistant strains of gram-negative aerobic bacilli. MRSA and multidrug-resistant gram-negative aerobic bacilli were simultaneously involved in five cases of PJIs. No statistically significant rising or decreasing linear trends were observed for multidrug-resistant bacteria, although, in the literature, a rising trend of PJIs caused by MDR strains is reported [31].

To be able to decide on an appropriate empirical antimicrobial therapy, the common microbiological causes of periprosthetic joint infections should be known, most importantly, at a local/regional scale, but also at a national/global scale. In our study, all cases of acute/acute hematogenous PJIs were caused by gram-positive aerobic or microaerophilic cocci pathogens. Both *Staphylococcus epidermidis* and methicillin-resistant *S. aureus* were involved in 91.66% of the acute/acute hematogenous PJIs cases. In this context, empirical antimicrobial therapy for acute PJIs should be focused on gram-positive aerobic or microaerophilic cocci. Similar data are also reported in the literature were most of the acute PJIs are caused by *Staphylococcus* spp. infection [27,28,30,31]

In previous reports in the literature, gram-negative aerobic bacilli were involved in <10% of cases of PJI [20,29], while Benito N, et al. report that gram-negative aerobic bacilli, mainly Enterobacterales, were isolated in 28% of PJIs [31]. Other studies report the frequency of these pathogens in PJIs that range from 17% up to 42% [27,32,33]. In our study, 35 strains (40.2%; 95% CI 29.9–50.6) of gram-negative aerobic bacilli were isolated; of them, Enterobacteriaceae comprised 19 (21.8; 13.8–31) strains and gram-negative nonfermenting bacilli comprised 16 (18.4; 10.3–26.4) strains. Our data did not confirm a rise of multidrug-resistant gram-negative infections as others have [31]. Multidrug-resistant gram-negative bacilli infections in clinical settings have steadily increased in the last years and are becoming a public health care issue of importance in Europe [34]. In our study, the following species accounted for 100% of all isolated strains of multidrug-resistant gram-negative aerobic bacilli: *Pseudomonas* spp., four strains; *Escherichia coli,* five strains; *Acinetobacter* spp., two strains; *Enterobacter cloacae* complex, three strains; and *Ralstonia pickettii*, three strains. 10 of 17 multidrug-resistant gram-negative aerobic bacilli strains that were isolated in our study were also extended-spectrum β-lactamases producing Enterobacterales strains. Concerning the resistance to specific antibiotics used in the management of PJIs, the most relevant is the resistance of some species at ciprofloxacin among gram-negative aerobic bacilli (11 strains). Resistance to quinolones is of greatest concern because ciprofloxacin is widely used in the treatment of PJIs caused by gram-negative bacilli [35]. Benito N et al. showed that almost 18% of gram-negative aerobic bacilli strains are resistant to quinolones and that there is an increasing resistance trend [31]. Fourteen strains of methicillin-resistant *S. aureus* and nine strains of methicillin-resistant *Staphylococcus epidermidis* were isolated in our study, in the study published by Benito N et. al, the percentage of MRSA increased from 4.7% in 2003–2004 to 9.5% in 2009–2010, and decreased to 7.6% in 2011–2012, a total number 180 strains of methicillin-resistant *S. aureus* being isolated (7.9%; 95% CI 6.7–9) [31].

Zeller V. et al. reported, based on the epidemiology profile of bone and joint infections from a French center over the last 12 years, that *Staphylococcus epidermidis* strains isolated from had a methicillin resistance of 84%; similar data to our study, where nine out of the eleven strains (81.81%) are methicillin-resistant strains [36]. According to EUCAST, it is known that vancomycin MIC values of 2 mg/L are on the border of the wild-type distribution and there may be an impaired clinical response. A MIC >2 mg/L for vancomycin measured using VITEK was not encountered, indicating that *Staphylococcus epidermidis* strains with reduced susceptibility to vancomycin were not isolated. We did not find an association between a MIC > 1 mg/L and co-resistance with rifampin; all rifampin-resistant strains (n = 2) had a vancomycin MIC <2 mg/L. For *Staphylococcus aureus* strains, a MIC >1 mg/L for teicoplanin measured using VITEK was not encountered. We did not find an association between a MIC >1 mg/L and co-resistance with rifampin; all isolated strains were susceptible to rifampin. Casenaz A et al., report that methicillin resistance was found in 15.2% (19/125) of *Staphylococcus* aureus strains and 49.3% (35/71) of CoNS strains. The authors also report that 29.1% of the infections were polymicrobial; in our study, 17.91% of cases were polymicrobial infections [37].

The current study also has some limitations. The main limitation of our study is the sample size of enrolled patients, which prevents us from reaching firm conclusions regarding the epidemiological trends. The study assesses microbial etiology and trends in one hospital, based on prospectively collected data. Nevertheless, the number of enrolled patients was sufficiently high to have an overview of the local possible etiologies of prosthetic joint infection. It is important to keep in mind that differences in patient characteristics, as well as in hospital and health care systems, means that our results cannot be generalized to other countries, and maybe not even to other regions in Romania. This was a monocentric, observational, retrospective cohort study. The center where this study was conducted was not a dedicated center for the treatment of PJIs. As with any culture study, a possibility exists that the isolated strains were secondary to contamination. Larger studies are needed to confirm our results; nevertheless, they are very promising. The use of a standardized definition of multidrug-resistant microorganisms is a strength of our study [38].

## 5. Conclusions

Most cases of periprosthetic joint infections in our study are monomicrobial, and *Staphylococcus aureus* is also the most common cause of infection in this study. A good understanding of the local epidemiology is necessary to optimize the treatment strategies of PJIs, and in our opinion, each center that treats PJIs should conduct regular epidemiological studies to optimize their empiric broad-spectrum antibiotic therapy. Based on our findings, empiric broad-spectrum antibiotic therapy in acute PJIs could be focused on the bacteria belonging to the gram-positive aerobic or microaerophilic cocci, but the results should be analyzed carefully, and the local resistance of the pathogens should always be taken into consideration. A total of four CoNS strains were resistant to gentamycin and to quinolones/fluoroquinolones. All strains of CoNS were susceptible to vancomycin, and trimethoprim/sulfamethoxazole; and four out of nineteen were resistant to quinolones/fluoroquinolones. We did not find any association between a MIC > 1 mg/L and co-resistance with rifampin. All strains of *Staphylococcus aureus* maintained their susceptibility to fluoroquinolones, trimethoprim/sulfamethoxazole, and to rifampicin. A MIC >1 mg/L for teicoplanin was not encountered. We did not find any association between a MIC >1 mg/L and co-resistance with rifampin; all isolated strains were susceptible to rifampin. All strains of *Enterococcus faecalis* maintained their susceptibility to vancomycin, and four out of the six were susceptible to trimethoprim/sulfamethoxazole. *Escherichia coli* strains, in relationship with piperacillin/ticarcillin, in combination with a beta-lactamase inhibitor (piperacillin/tazobactam; ticarcillin/clavulanic acid), preserved the sensitivity all strains. As for susceptibility to carbapenems, it was preserved in all three tested antibiotics. Thus, two of the most frequently used classes of antibiotics in the management of infections associated with orthopedic implants remain under discussion; five strains being sensitive to ciprofloxacin, five to trimethoprim/sulfamethoxazole, and all strains being intermediately sensitive to norfloxacin. Susceptibility to fosfomycin of *Escherichia coli* is preserved. For the *Enterobacter* species isolated strains, five of the isolated strains were susceptible to Piperacillin/tazobactam, 50% were resistant to cephalosporins, 75% were resistant to gentamicin, and all strains maintained their susceptibility to trimethoprim/sulfamethoxazole. For the *Pseudomonas*-species isolated strains (n = 9), five of the isolated strains were susceptible to ticarcillin/clavulanic acid, four to piperacillin/tazobactam, five to ceftazidime, six to imipenem, seven to meropenem, seven to ceftazidime/avibactam and ceftolozane/tazobactam, and five to ciprofloxacin. All strains of *Ralstonia picketti* maintained their susceptibility to piperacillin, imipenem, meropenem, ciprofloxacin, pefloxacin, minocycline, and trimethoprim/sulfamethoxazole. Our data shows that it is also important to optimize and improve the antimicrobial treatment strategies based on the local AST data that involve antibiotics which have activity against biofilm-related infections. Multidisciplinary teams and accurate etiological diagnosis are necessary for the management of periprosthetic joint infection cases.

## Figures and Tables

**Figure 1 microorganisms-11-00116-f001:**
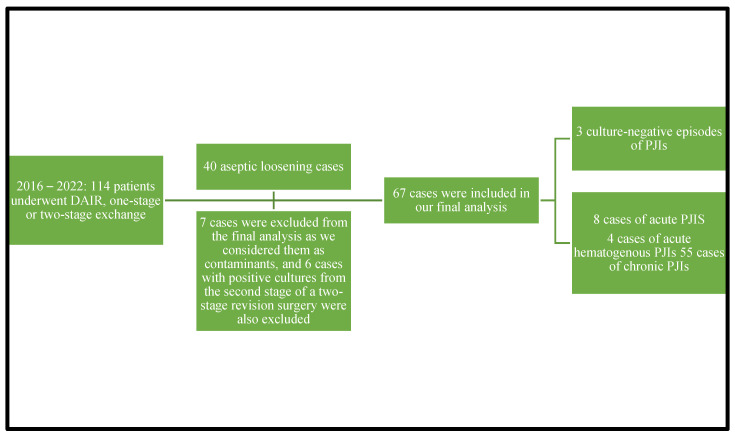
Flow diagram showing details of the enrolled patients.

**Figure 2 microorganisms-11-00116-f002:**
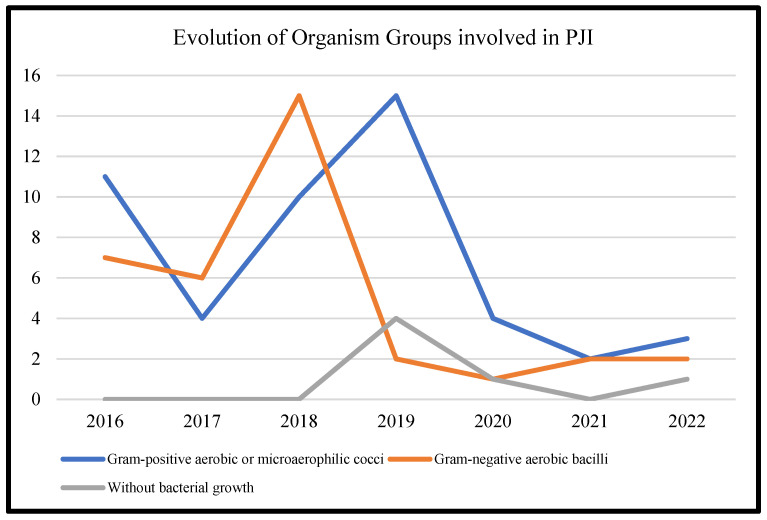
Trends in the microbial etiology of PJIs.

**Figure 3 microorganisms-11-00116-f003:**
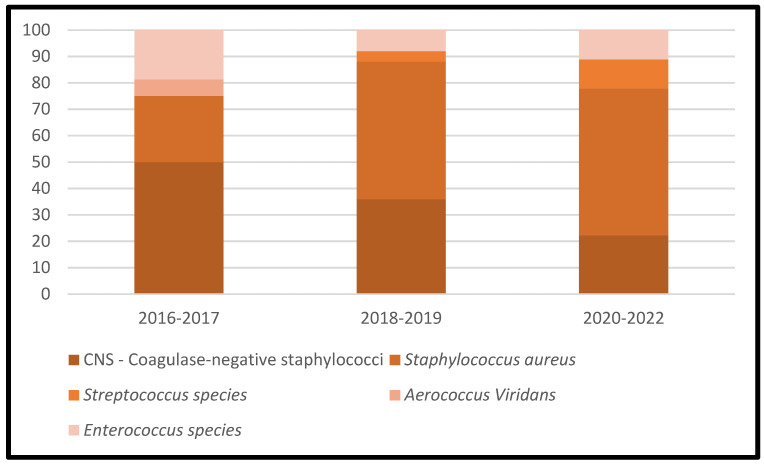
Trends in the microbial etiology of PJIs: distribution of gram-positive aerobic or microaerophilic cocci.

**Figure 4 microorganisms-11-00116-f004:**
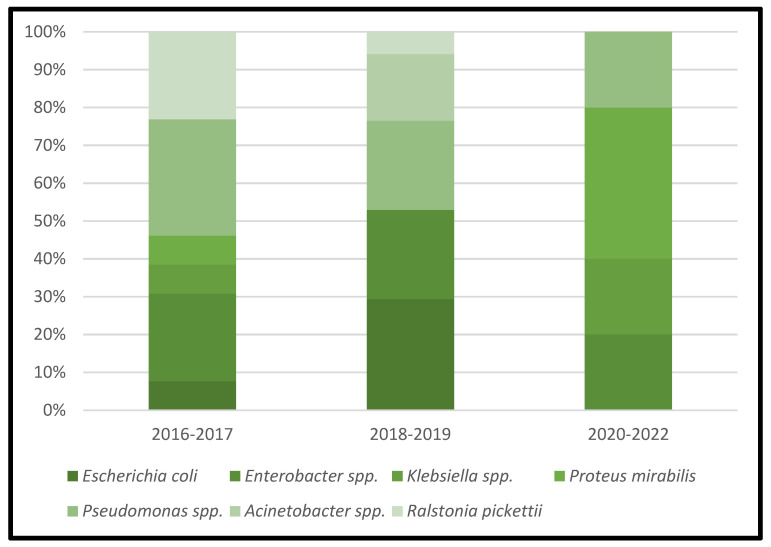
Trends in the microbial etiology of PJIs: distribution of gram-negative aerobic bacilli.

**Figure 5 microorganisms-11-00116-f005:**
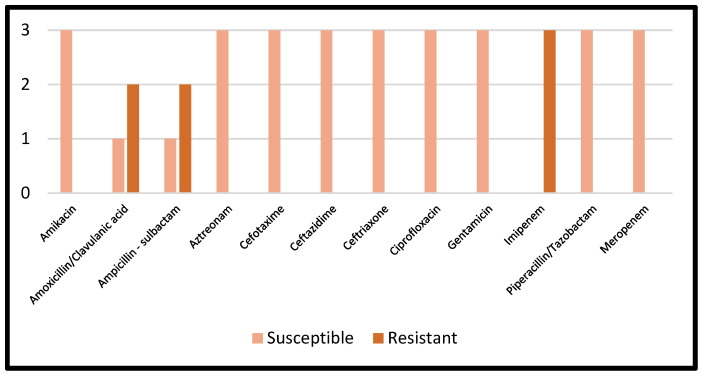
AST results of *Proteus mirabilis*.

**Table 1 microorganisms-11-00116-t001:** 2021 European Bone and Joint Infection Society (EBJIS) definition for the diagnosis of periprosthetic joint infection [26].

	Infection Unlikely	Infection Likely	Infection Confirmed
	(All Findings Negative)	(Two Positive Findings) ^a^	(Any Positive Finding)
**Clinical and blood workup**			
Clinical features	Clear alternative reason for implant dysfunction (e.g., fracture, implant breakage, malposition, tumour)	(1) Radiological signs of loosening within the first five years after implantation	Sinus tract with evidence of communication to the joint or visualization of the prosthesis
		(2) Previous wound healing problems	
		(3) History of recent fever or bacteraemia	
		(4) Purulence around the prosthesis ^b^	
C-reactive protein		>10 mg/L (1 mg/dL) ^c^	
**Synovial fluid cytological analysis ^d^**			
Leukocyte count ^c^ (cells/μL)	≤1500	>1500	>3000
**PMN (%) ^c^**	≤65%	>65%	>80%
**Synovial fluid biomarkers**			
Alpha-defensin ^e^			Positive immunoassay or lateral-flow assay^e^
**Microbiology ^f^**			
Aspiration fluid		Positive culture	
**Intraoperative (fluid and tissue)**	All cultures negative	Single positive culture ^g^	≥two positive samples with the same microorganism
**Sonication ^h^ (CFU/mL)**	No growth	>1 CFU/mL of any organism ^g^	>50 CFU/mL of any organism
**Histology ^c,i^**			
High-power field (400× magnification)	Negative	Presence of ≥five neutrophils in a single HPF	Presence of ≥ five neutrophils in ≥five HPF
			Presence of visible microorganisms
**Others**			
Nuclear imaging	Negative three-phase isotope bone scan^c^	Positive WBC scintigraphy ^j^	

^a^ Infection is only likely if there is a positive clinical feature or raised serum C-reactive protein (CRP), together with another positive test (synovial fluid, microbiology, histology, or nuclear imaging). ^b^ Except in adverse local tissue reaction (ALTR) and crystal arthropathy cases. ^c^ Should be interpreted with caution when other possible causes of inflammation are present: gout or other crystal arthropathy, metallosis, active inflammatory joint disease (e.g., rheumatoid arthritis), periprosthetic fracture, or the early postoperative period. ^d^ These values are valid for hips and knee periprosthetic joint infection (PJI). Parameters are only valid when clear fluid is obtained and no lavage has been performed. Volume for the analysis should be > 250 μL, ideally 1 mL, collected in an EDTA containing tube and analyzed in <1 h, preferentially using automated techniques. For viscous samples, pre-treatment with hyaluronidase improves the accuracy of optical or automated techniques. In case of bloody samples, the adjusted synovial WBC = synovial WBC observed—[WBC blood/RBC blood × RBC synovial fluid] should be used. ^e^ Not valid in cases of ALTR, haematomas, or acute inflammatory arthritis or gout. ^f^ If antibiotic treatment has been given (not simple prophylaxis), the results of microbiological analysis may be compromised. In these cases, molecular techniques may have a place. Results of culture may be obtained from preoperative synovial aspiration, preoperative synovial biopsies or (preferred) from intraoperative tissue samples. ^g^ Interpretation of single positive culture (or <50 UFC/mL in sonication fluid) must be cautious and taken together with other evidence. If a preoperative aspiration identified the same microorganism, they should be considered as two positive confirmatory samples. Uncommon contaminants or virulent organisms (e.g., *Staphylococcus aureus* or Gram negative rods) are more likely to represent infection than common contaminants (such as coagulase-negative staphylococci, micrococci, or *Cutibacterium acnes*). ^h^ If centrifugation is applied, then the suggested cut-off is 200 CFU/mL to confirm infection. If other variations to the protocol are used, the published cut-offs for each protocol must be applied. ^i^ Histological analysis may be from preoperative biopsy, intraoperative tissue samples with either paraffin, or frozen section preparation. ^j^ WBC scintigraphy is regarded as positive if the uptake is increased at the 20-h scan, compared to the earlier scans (especially when combined with complementary bone marrow scan).

**Table 2 microorganisms-11-00116-t002:** Baseline characteristics of the enrolled patients diagnosed with PJI from 2016 through 2022.

Characteristic	No. of Cases (*n* = 67)
Median age (interquartile range, standard deviation), years	68.5 (12, ±10.88)
Male gender	34 (50.74)
Comorbidities	
Any comorbidity	62 (92.53)
Diabetes mellitus	28 (41.79)
Arterial hypertension	55 (82.08)
Chronic heart disease	24 (35.82)
Heart failure	31 (46.26)
Ischemic stroke	11 (16.41)
Peripheral vascular disease	27 (40.29)
Chronic obstructive pulmonary disease	6 (8.95)
Cancer	4 (5.97)
Neurological disease	5 (7.42)
Chronic kidney disease	7 (10.44)
Connective tissue disease	2 (2.98)
Liver disease	10 (14.92)
Rheumatoid arthritis	3 (4.47)
Obesity	26 (38.80)
Charlson Comorbidity Index, median (interquartile range)	3 (2)
Index arthroplasty site	
Total hip arthroplasty	38
Hip hemiarthroplasty	2
Total knee arthroplasty	27
ASA score, median (interquartile range)	2 (1)
Indication for index arthroplasty	
osteoarthritis	37 (55.22)
rheumatoid arthritis	2 (2.98)
femoral neck fracture	7 (10.44)
other	21 (31.34)

If not otherwise stated, data are no. (%) of patients with indicated item.

**Table 3 microorganisms-11-00116-t003:** Lists of causative microorganisms of PJIs during the study period (2016 through 2021).

Microorganism or Group	Total No. of Positive Cultures n (%; 95% CI)
**Gram-positive aerobic or microaerophilic cocci**	48 (55.2; 44.8–65.5)
**CNS—Coagulase-negative *staphylococci***	19 (21.8; 13.8–31)
*Staphylococcus epidermidis*	11 (12.6; 5.8–20.7)
*Staphylococcus lentus*	3 (3.4; 0–6.9)
*Staphylococcus xylosus*	2 (2.3; 0–5.7)
*Staphylococcus hominis*	2 (2.3; 0–5.7)
*Staphylococcus haemolyticus*	1 (1.1; 0–3.4)
** *Staphylococcus aureus* **	21 (24.1; 14.9–33.3)
Methicillin-resistant *S. aureus*	14 (16.1; 9.2–24.1)
Methicillin-susceptible *S. aureus*	7 (8; 3.4–13.8)
***Streptococcus* species**	2 (2.3; 0–5.7)
*Streptococcus* group D	2 (2.3; 0–5.7)
** *Aerococcus Viridans* **	1 (1.1; 0–3.4)
***Enterococcus* species**	6 (6.9; 2.3–12.6)
*Enterococcus faecalis*	6 (6.9; 2.3–12.6)
**Gram-negative aerobic bacilli**	35 (40.2; 29.9–50.6)
**Enterobacterales**	19 (21.8; 13.8–31)
*Escherichia coli*	6 (6.9; 2.3–12.6)
*Enterobacter* spp.	**8 (9.2; 3.4–14.9)**
*Enterobacter cloacae* complex	7 (8; 3.4–13.8)
*Enterobacter amnigenus* 2	1 (1.1; 0–3.4)
*Klebsiella* spp.	2 (2.3; 0–5.7)
*Proteus mirabilis*	3 (3.4; 0–8)
**Gram-negative nonfermenting bacilli**	16 (18.4; 10.3–26.4)
*Pseudomonas* spp.	9 (10.3; 3.4–16.1)
*Pseudomonas fluorescens*	1 (1.1; 0–3.4)
*Pseudomonas aeruginosa*	8 (9.2; 3.4–14.9)
*Acinetobacter spp.*	3 (3.4; 0–8)
*Ralstonia pickettii*	4 (4.6; 1.1–9.2)
**Without bacterial growth**	3 (3.4; 0–8)

**Table 4 microorganisms-11-00116-t004:** MICs (mg/L) of antimicrobials for *Coagulase-negative staphylococci* isolates.

Bacterial Strain/Antibiotic (MIC/Result)	Cefoxitin Screening	Benzylpenicillin		Oxacillin		Imipenem		Genta-mycin		Cipro-floxacin		Moxifloxacin		Erythro-mycin		Clinda-mycin		Linezolid		Teico-planin		Vanco-mycin		Tetra-cycline		Tigecycline		Fosfomycin		Fusific Acid		Rifampicin		Trimethropim/ Sulfamethoxazole	
***S. epidermidis* 1**	Neg	0.03	S	0.25	S	1	S	8	R	0.5	S	0.25	S	0.25	S	0.25	S	1	S	0.5	S	0.5	S	2	R	0.12	S	8	S	8	S	0.5	I	10	S
***S. epidermidis* 2**	Pos	0.25	R	4	R	1	S	4	R	0.5	S	0.25	S	0.25	S	0.25	S	2	S	4	S	1	S	2	R	0.25	S	6	S	0.5	S	0.5	I	10	S
***S. epidermidis* 3**	Neg	0.5	R	4	R	1	S	4	R	8	R	2	R	8	R	0.25	R	1	S	4	S	1	S	2	R	0.12	S	6	S	16	R	0.5	I	10	S
***S. epidermidis* 4**	Pos	0.5	R	4	R	1	S	4	R	0.5	S	0.25	S	8	R	0.25	S	1	S	4	S	1	S	2	R	0.25	S	6	S	16	R	0.25	S	10	S
***S. epidermidis* 5**	Neg	0.25	R	4	R	1	S	4	R	0.5	S	0.25	S	0.25	S	0.25	S	1	S	2	S	1	S	2	R	0.25	S	6	S	0.5	S	0.25	S	10	S
***S. epidermidis* 6**	Pos	0.25	R	4	R	1	S	4	R	0.5	S	0.25	S	0.25	S	0.25	S	1	S	2	S	1	S	2	R	0.25	S	6	S	0.5	S	0.25	S	10	S
***S. epidermidis* 7**	Neg	0.5	R	4	R	1	S	8	R	8	R	8	R	8	R	0.5	R	1	S	1	S	1	S	2	R	0.12	S	64	S	0.5	S	1	I	10	S
***S. epidermidis* 8**	Pos	0.03	S	0.25	S	1	S	8	R	0.5	S	0.25	S	0.25	S	0.25	S	1	S	0.5	S	0.5	S	2	R	0.12	S	8	S	8	S	0.5	I	10	S
***S. epidermidis* 9**	Neg	0.25	R	4	R	1	S	4	R	0.5	S	0.25	S	0.25	S	0.25	S	2	S	4	S	1	S	2	R	0.25	S	6	S	0.5	S	0.5	I	10	S
***S. epidermidis* 10**	Pos	0.5	R	4	R	1	S	4	R	8	R	2	R	8	R	0.25	R	1	S	4	S	1	S	2	R	0.12	S	6	S	16	R	0.5	I	10	S
***S. epidermidis* 11**	Neg	0.5	R	4	R	1	S	4	R	0.5	S	0.25	S	8	R	0.25	S	1	S	4	S	1	S	2	R	0.12	S	6	S	16	R	0.25	S	10	S
***S. lentus* 1**	Pos	0.5	R	4	R	2	R	0.5	S	0.5	S	0.25	S	8	R	0.25	S	2	S	1	S	1	S	16	R	0.12	S	8	S	0.5	S	32	R	10	S
***S. lentus* 2**	Neg	>0.5	R	>4	R	2	R	<0.5	S	<0.5	S	<0.25	S	>8	R	<0.25	S	2	S	1	S	1	S	>16	S	<0.12	S	4	S	<0.5	S	>32	R	<10	S
***S. xylosus* 1**	Pos	0.12	S	0.5	S	1	S	0.5	S	0.5	S	0.25	S	>8	R	0.5	S	2	S	2	S	1	S	<1	S	<0.12	S	8	S	2	R	0.5	S	10	S
***S. xylosus* 2**	Neg	0.12	S	0.5	S	1	S	0.5	S	0.5	S	0.25	S	>8	R	0.5	S	2	S	2	S	1	S	<1	S	<0.12	S	8	S	2	R	0.5	S	10	S
***S. xylosus* 3**	Pos	0.12	S	0.5	S	1	S	0.5	S	0.5	S	0.25	S	>8	R	0.5	S	2	S	2	S	1	S	<1	S	<0.12	S	8	S	2	R	0.5	S	10	S
*S. haemoliticus*	Neg	0.5	R	4	R	2	R	8	R	0.5	S	0.25	S	>8	R	0.5	S	2	S	2	S	1	S	16	R	<0.12	S	32	R	2	R	1	I	10	S
***S. hominis* 1**	Pos	0.5	R	4	R	2	R	0.5	S	0.5	S	0.25	S	>8	R	0.5	S	2	S	2	S	1	S	16	R	<0.12	S	8	S	1	R	0.5	S	10	S
***S. hominis* 2**	Neg	0.5	R	4	R	2	R	1	S	8	R	2	R	>8	R	1	R	2	S	2	S	1	S	16	R	<0.12	S	32	R	0.5	S	0.5	S	10	S

**Table 5 microorganisms-11-00116-t005:** MICs (mg/L) of antimicrobials for *Staphylococcus aureus* isolates.

Bacterial Strain/Antibiotic (MIC/Result)	Cefoxitin		Ciprofloxacin		Clindamicin		Chloram-phenicol		Trimethropim/ Sulfamethoxazole		Erythromycin		Gentamicin		Oxacillin		Penicillin		Rifampin		Tetracycline		Linezolid		Teicoplanin	
* **S. aureus 1** *	64	R	0.5	S	0.5	R	4	S	1	S	4	R	1	S	4	R	64	R	0.016	S	32	R	4	S	1	S
* **S. aureus 2** *	64	R	0.5	S	0.5	R	4	S	1	S	4	R	1	S	4	R	64	R	0.016	S	32	R	4	S	1	S
* **S. aureus 3** *	64	R	0.5	S	0.5	R	4	S	1	S	4	R	1	S	4	R	64	R	0.016	S	32	R	4	S	1	S
* **S. aureus 4** *	64	R	0.5	S	0.5	R	4	S	1	S	4	R	1	S	4	R	64	R	0.016	S	32	R	4	S	1	S
* **S. aureus 5** *	64	R	0.5	S	0.5	R	4	S	1	S	4	R	1	S	4	R	64	R	0.016	S	32	R	4	S	1	S
* **S. aureus 6** *	64	R	0.5	S	0.5	R	4	S	1	S	4	R	1	S	4	R	64	R	0.016	S	32	R	4	S	1	S
* **S. aureus 7** *	64	R	0.5	S	0.125	S	4	S	1	S	1	S	1	S	4	R	64	R	0.016	S	32	R	4	S	1	S
* **S. aureus 8** *	64	R	0.5	S	0.5	R	4	S	1	S	1	S	1	S	4	R	64	R	0.016	S	32	R	4	S	1	S
* **S. aureus 9** *	64	R	0.5	S	0.5	R	4	S	1	S	4	R	1	S	4	R	64	R	0.016	S	32	R	4	S	1	S
* **S. aureus 10** *	64	R	0.5	S	0.5	R	4	S	1	S	4	R	1	S	4	R	64	R	0.016	S	32	R	4	S	1	S
* **S. aureus 11** *	64	R	0.5	S	0.5	R	4	S	1	S	4	R	1	S	4	R	64	R	0.016	S	32	R	4	S	1	S
* **S. aureus 12** *	64	R	0.5	S	0.5	R	4	S	1	S	4	R	8	R	4	R	64	R	0.016	S	32	R	4	S	1	S
* **S. aureus 13** *	64	R	0.5	S	0.125	S	4	S	1	S	1	S	1	S	4	R	64	R	0.016	S	32	R	4	S	1	S
* **S. aureus 14** *	64	R	0.5	S	0.5	R	4	S	1	S	4	R	8	R	4	R	64	R	0.016	S	32	R	4	S	1	S
* **S. aureus 15** *	2	S	0.5	S	0.5	R	4	S	1	S	4	R	1	S	0.125	S	64	R	0.016	S	32	R	4	S	1	S
* **S. aureus 16** *	2	S	0.5	S	0.5	R	4	S	1	S	4	R	1	S	0.125	S	64	R	0.016	S	32	R	4	S	1	S
* **S. aureus 17** *	2	S	0.5	S	0.5	R	4	S	1	S	4	R	2	R	0.125	S	64	R	0.016	S	32	R	4	S	1	S
* **S. aureus 18** *	2	S	0.5	S	0.5	R	4	S	1	S	4	R	1	S	0.125	S	64	R	0.016	S	32	R	4	S	1	S
* **S. aureus 19** *	2	S	0.5	S	0.5	R	4	S	1	S	4	R	1	S	0.125	S	64	R	0.016	S	32	R	4	S	1	S
* **S. aureus 20** *	2	S	0.5	S	0.5	R	4	S	1	S	4	R	1	S	0.125	S	64	R	0.016	S	32	R	4	S	1	S
* **S. aureus 21** *	2	S	0.5	S	0.5	R	4	S	1	S	4	R	1	S	0.125	S	64	R	0.016	S	32	R	4	S	1	S

**Table 6 microorganisms-11-00116-t006:** MICs (mg/L) of antimicrobials for *Enterococcus faecalis* isolates.

Bacterial Strain/Antibiotic (MIC/Result)	Ampicilin		Imipenem		Gentamicin High Level		Streptomycin High Level		Cipro-floxacin		Moxi-floxacin *		Erythro-micin		Clinda-mycin		Linezolid		Teicoplanin		Vanco-mycin		Tetracyclin		Tigecycline		Trimethoprim/Sulfamethoxazole *	
*E. faecalis* 1	2	S	1	S	SYN-R	R	SYN-R	S	8	R	4	R	8	R	8	R	2	S	0.5	S	1	S	16	R	0.12	S	20	I
*E. faecalis* 2	2	S	1	S	SYN-R	R	SYN-R	S	4	S	1	S	8	R	8	R	2	S	0.5	S	1	S	16	R	0.12	S	0.064	S
*E. faecalis* 3	2	S	1	S	SYN-R	R	SYN-R	R	8	R	4	R	8	R	8	R	2	S	0.5	S	1	S	16	R	0.12	S	20	I
*E. faecalis* 4	2	S	1	S	SYN-R	R	SYN-R	R	16	R	1	S	8	R	8	R	2	S	0.5	S	1	S	16	R	0.12	S	0.064	S
*E. faecalis* 5	2	S	1	S	SYN-R	S	SYN-R	R	4	S	1	S	8	R	8	R	2	S	0.5	S	1	S	16	R	0.12	S	0.064	S
*E. faecalis* 6	2	S	1	S	SYN-R	S	SYN-R	S	4	S	1	S	8	R	8	R	2	S	0.5	S	1	S	16	R	0.12	S	0.064	S

* MIC ECOFF.

**Table 7 microorganisms-11-00116-t007:** MICs (mg/L) of antimicrobials for *Escherichia coli* isolates.

Bacterial Strain/Antibiotic (MIC/Result)	Ticarcilina/Clavulanic Acid		Piperacillin/Tazobactam		Cefotaxime		Ceftazidime		Cefepime		Ertapenem		Imipenem		Meropenem		Amikacina		Gentamicina		Ciprofloxacin		Norfloxacin		Fosfomycin		Nitrofurantoin		Trimethoprim/Sulfamethoxazole	
* **E. Coli 1** *	8	S	<4	S	32	R	16	R	32	R	<0.5	S	1	S	<0.25	S	<2	S	32	R	0.5	S	0.125	I	0.5	S	64	R	0.06	S
* **E. Coli 2** *	8	S	<4	S	1	S	1	S	2	S	<0.5	S	1	S	<0.25	S	<2	S	32	R	2	R	0.125	I	0.5	S	64	R	0.06	S
* **E. Coli 3** *	8	S	<4	S	32	R	16	R	32	R	<0.5	S	1	S	<0.25	S	<2	S	32	R	1	S	0.125	I	0.5	S	64	R	64	R
* **E. Coli 4** *	8	S	<4	S	0.06	S	0.25	S	0.06	S	<0.5	S	1	S	<0.25	S	<2	S	0.5	S	1	S	0.125	I	0.5	S	64	R	0.06	S
* **E. Coli 5** *	8	S	<4	S	32	R	16	R	32	R	<0.5	S	1	S	<0.25	S	<2	S	32	R	2	R	0.125	I	0.5	S	64	R	0.125	S
* **E. Coli 6** *	8	S	<4	S	32	R	16	R	32	R	<0.5	S	1	S	<0.25	S	<2	S	32	R	1	S	0.125	I	0.5	S	64	R	0.06	S

**Table 8 microorganisms-11-00116-t008:** MICs (mg/L) of antimicrobials for *Enterobacter* spp. isolates.

Bacterial Strain/Antibiotic (MIC/Result)	Amoxicillin/Clavulanic Acid		Piperacillin/Tazobactam		Cefotaxime		Ceftazidime		Cefepime		Ertapenem		Imipenem		Meropenem		Amikacina		Gentamicina		Ciprofloxacin		Norfloxacin		Fosfomycin		Trimethoprim/Sulfamethoxazole	
*Enterobacter amnigenus* 2	>32	R	<4	S	>64	R	16	R	32	R	<0.5	S	1	S	<0.25	S	<2	S	32	R	2	I	8	I	64	S	0.125	S
*E. cloacae complex* 1	4	R	<4	S	<1	S	<1	S	<1	S	<0.5	S	<0.25	S	<0.25	S	<2	S	1	S	<0.25	S	<0.5	S	128	I	0.125	S
*E. cloacae complex* 2	4	R	<4	S	<1	S	0.125	S	<1	S	<0.5	S	<0.25	S	0.06	S	16	I	64	R	8	R	<0.5	S	128	I	0.125	S
*E. cloacae complex* 3	4	R	<4	S	32	R	16	R	32	R	<0.5	S	<0.25	S	<0.25	S	16	I	32	R	8	R	<0.5	S	128	I	0.125	S
*E. cloacae complex* 4	4	R	32	R	16	R	16	R	32	R	<0.5	S	2	R	2	R	<2	S	32	R	8	S	16	R	64	S	0.125	S
*E. cloacae complex* 5	4	R	8	I	16	R	32	R	32	R	<0.5	S	<0.25	S	<0.25	S	<2	S	32	R	8	R	16	R	128	I	0.125	S
*E. cloacae complex* 6	4	R	32	R	<1	S	<1	S	<1	S	<0.5	S	<0.25	S	<0.25	S	<2	S	1	S	<0.25	S	<0.5	S	128	I	0.125	S
*E. cloacae complex* 7	4	R	<4	S	<1	S	<1	S	<1	S	<0.5	S	<0.25	S	<0.25	S	<2	S	32	R	<0.25	S	<0.5	S	128	I	0.125	S

**Table 9 microorganisms-11-00116-t009:** MICs (mg/L) of antimicrobials for *Pseudomonas* spp. isolates.

Bacterial Strain/Antibiotic (MIC/Result)	Amikacin		Ticarcillin/Clavulanic Acid		Piperacillin/Tazobactam		Ceftazidime		Imipenem		Meropenem		Ceftazidime/Avibactam Sensibil		Ceftolozane/Tazobactam		Ciprofloxacin	
*Pseudomonas aeruginosa 1*	4	S	8	S	4	S	2	S	0.5	S	0.5	S	1	S	1	S	0.25	S
*Pseudomonas aeruginosa 2*	64	R	32	R	32	I	128	R	16	R	2	S	8	S	4	S	8	R
*Pseudomonas aeruginosa 3*	4	S	8	S	8	S	2	S	0.5	S	0.5	S	1	S	1	S	0.5	S
*Pseudomonas aeruginosa 4*	32	R	32	R	64	R	128	R	16	R	4	R	64	R	64	R	8	R
*Pseudomonas aeruginosa 5*	4	S	8	S	8	S	2	S	0.5	S	0.5	S	0.5	S	1	S	0.5	S
*Pseudomonas aeruginosa 6*	32	R	32	R	64	R	128	R	8	R	4	R	64	R	32	R	8	R
*Pseudomonas aeruginosa 7*	4	S	16	I	32	I	2	S	0.5	S	0.5	S	1	S	1	S	0.5	S
*Pseudomonas aeruginosa 8*	4	S	4	S	8	S	2	S	0.5	S	0.5	S	0.5	S	1	S	0.5	S
*Pseudomonas fluorescens*	32	R	6	S	128	R	64	R	0.5	S	0.5	S	1	S	1	S	4	R

**Table 10 microorganisms-11-00116-t010:** MICs (mg/L) of antimicrobials for *Acinetobacter* spp. isolates.

Bacterial Strain/Antibiotic (MIC/Result)	Minocycline		Colistin		Cefazolin		Cefepime		Ceftazidime		Ciprofloxacin		Ceftriaxone		Gentamicin		Meropenem		Levofloxacin		Piperacillin		Tobramycin		Trimethoprim/Sulfamethoxazole	
*Acinetobacter spp.1*		S	1	S	64	R	1	S	0.25	S	0.5	S	4	S	2	S	2	S	1	S	16	S	4	S	0.5	S
*Acinetobacter spp.2*		R	1	S	64	R	16	R	32	I	4	R	4	S	2	S	0.5	S	1	S	64	R	4	S	0.5	S
*Acinetobacter spp.3*		I	1	S	64	R	16	R	0.25	S	4	R	4	S	4	R	8	R	4	R	64	R	16	R	2	R

**Table 11 microorganisms-11-00116-t011:** MICs (mg/L) of antimicrobials for *Ralstonia picketti* isolates.

Bacterial Strain/Antibiotic (MIC/Result)	Ticarcillin		Piperacillin		Piperacillin/Tazobactam		Ceftazidime		Cefepime		Aztreonam		Imipenem		Meropenem		Amikacin		Genatmicin		Tobramycin		Ciprofloxacin		Pefloxacin		Minocycline		Colistin		Trimethropim/Sulfamethoxazole	
*Ralstonia picketti* 1	16	S	4	S	16	S	16	I	64	R	64	R	0.25	S	4	S	64	R	16	R	16	R	0.25	S	1	S	1	S	8	R	20	S
*Ralstonia picketti 2*	128	R	4	S	16	S	16	I	1	S	64	R	0.25	S	4	S	64	R	4	S	16	R	0.25	S	1	S	1	S	16	R	20	S
*Ralstonia picketti 3*	128	R	4	S	16	S	16	I	1	S	64	R	0.25	S	4	S	64	R	4	S	16	R	0.25	S	1	S	1	S	16	R	20	S
*Ralstonia picketti 4*	64	I	<4	S	16	S	8	S	<1	S	>64	R	0.5	S	2	S	8	S	8	I	4	S	0.125	S	0.5	S	<1	S	16	R	20	S

## Data Availability

All data generated or analyzed during this study are included in this published article.

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
