# Peer review of "Are There Any Changes in the Causative Microorganisms Isolated in the Last Years from Hip and Knee Periprosthetic Joint Infections? Antimicrobial Susceptibility Test Results Analysis"

_microorganisms, 2023, doi:10.3390/microorganisms11010116_

Round 1
Reviewer 1 Report
The paper by Fleaca and colleagues addresses the pivotal argument of joint infections. The manuscript is well written, protocols and data scientifically sounds and are presented in a clear way. Nonetheless, there are some minor modifications that could improve their work quality.
The paper was submitted to a "biofilm" section, but biofilm is only mentioned at the beginning of the introduction. Please, consider to add more information about biofilm role in joint infections and its antibiotic resistance/treatment.
Conclusions are too much epidemiologically-oriented. Although the aim of the paper, according to the title, is to evaluate the changes in the microorganisms causing joint infections, data collected about their antibiotic resistance are almost neglected in the discussion. Likewise, the reader expects a more punctual microbiological analysis, at least for the main causative agents (see for example Stracquadanio S, Musso N, Costantino A, Lazzaro LM, Stefani S, Bongiorno D. Staphylococcus aureus Internalization in Osteoblast Cells: Mechanisms, Interactions and Biochemical Processes. What Did We Learn from Experimental Models? Pathogens. 2021 Feb 19;10(2):239. doi: 10.3390/pathogens10020239.)
Minor comments:
- Lines 33, 208, 226, 347, 384, 393 Enterobacteriaceae should be replaced with Enterobacterales
- Line 34, ones could be removed at the end of the sentence
- Lines 76 (and 97), which media did the authors use?
- Lines 81-82, the sentence is not clear, please rephrase it
- Line 89, otherS
- Lines 90-91, aÈ™ the authors used 50mL of the fluid in the further passages, how much saline they put?
- Line 111, in some articles is not necessary
- Line 121, the full stop after positive must be removed
- Lines 139-140, specie must be italicized
- Line 164, p-value usually is </= not only <
- Line 184, "as already published []" adding the reference(s)
- Line 190, table no.2 not no.1
- Line 193, what does 3.92.53% mean?
- Table 2, Enterobacteriaceae should be replaced with Enterobacterales
- Line 204, authors should add a brief sentence at the beginning of the paragraph (According to our results, 48...)
- Lines 213-214, no... not, check the sentence
- Line 355, again, there is a review addressing the important role of S. aureus in these infections (Stracquadanio S, Musso N, Costantino A, Lazzaro LM, Stefani S, Bongiorno D. Staphylococcus aureus Internalization in Osteoblast Cells: Mechanisms, Interactions and Biochemical Processes. What Did We Learn from Experimental Models? Pathogens. 2021 Feb 19;10(2):239. doi: 10.3390/pathogens10020239.)
Author Response
Sibiu, 24.12.2022
To
the Editors of Microorganisms®
Dear Editor-in-Chief,
Dear Editor,
Dear reviewer,
Thank you for reviewing our manuscript. Please find attached a revised version of our manuscript, “Are there any changes in the causative microorganisms isolated in the last years from hip and knee periprosthetic joint infections?”.
Your and the reviewers’ comments were highly insightful and enabled us to greatly improve the quality of our manuscript. We have modified the manuscript in response to the comments. Attached is our point-by-point response to each comment.
Reviewer Comments:
Reviewer 1
The paper by Fleaca and colleagues addresses the pivotal argument of joint infections. The manuscript is well written, protocols and data scientifically sounds and are presented in a clear way. Nonetheless, there are some minor modifications that could improve their work quality.
The paper was submitted to a "biofilm" section, but biofilm is only mentioned at the beginning of the introduction. Please, consider to add more information about biofilm role in joint infections and its antibiotic resistance/treatment.
Conclusions are too much epidemiologically-oriented. Although the aim of the paper, according to the title, is to evaluate the changes in the microorganisms causing joint infections, data collected about their antibiotic resistance are almost neglected in the discussion. Likewise, the reader expects a more punctual microbiological analysis, at least for the main causative agents (see for example Stracquadanio S, Musso N, Costantino A, Lazzaro LM, Stefani S, Bongiorno D. Staphylococcus aureus Internalization in Osteoblast Cells: Mechanisms, Interactions and Biochemical Processes. What Did We Learn from Experimental Models? Pathogens. 2021 Feb 19;10(2):239. doi: 10.3390/pathogens10020239.)
Answer: Thank you for taking your precious time to be able to assess our manuscript. The comments were highly insightful and enabled us to improve our manuscript. We added some more data regarding biofilm and its role in infections and also in the AST. We also added some data regarding AST in the conclusions section of the manuscript. Thank you also for the suggested reference.
Minor comments:
- Lines 33, 208, 226, 347, 384, 393 Enterobacteriaceae should be replaced with Enterobacterales
A: Thank you for the suggestion. Done!
- Line 34, ones could be removed at the end of the sentence
A: Thank you for the suggestion. Done!
- Lines 76 (and 97), which media did the authors use?
A: Thank you for the suggestion. We added the types of media that were used.
- Lines 81-82, the sentence is not clear, please rephrase it
A: We rephrased the sentence, hope that it is clear now.
- Line 89, otherS
A: Thank you for the suggestion.
- Lines 90-91, aÈ™ the authors used 50mL of the fluid in the further passages, how much saline they put?
A: Thank you for the question, and we hope that with the following answer we addressed the issue. Having in mind that the smallest transport container has 0.52 ltr. and that at least ¾ of the implant was be covered with sterile Ringer's or saline solution around 200-300ml of solution was added, unfortunately an exact quantity that was put in each container was not recorded.
- Line 111, in some articles is not necessary
A: Done!
- Line 121, the full stop after positive must be removed
A: Done! Thank you for pointing this also.
- Lines 139-140, specie must be italicized
A: Done! Thank you for pointing this also.
- Line 164, p-value usually is </= not only <
A: Done! Thank you for pointing this also.
- Line 184, "as already published []" adding the reference(s)
A: Done!
- Line 190, table no.2 not no.1
A: Done! Thank you for pointing this also. We greatly appreciate your time dedicated to perform our manuscript review.
- Line 193, what does 3.92.53% mean?
A: We rephrased the sentence, hope that it is clear now.
- Table 2, Enterobacteriaceae should be replaced with Enterobacterales
A: Done!
- Line 204, authors should add a brief sentence at the beginning of the paragraph (According to our results, 48...)
A: Done! Thank you for pointing this also.
- Lines 213-214, no... not, check the sentence
A: Done! Thank you for pointing this also.
- Line 355, again, there is a review addressing the important role of S. aureus in these infections (Stracquadanio S, Musso N, Costantino A, Lazzaro LM, Stefani S, Bongiorno D. Staphylococcus aureus Internalization in Osteoblast Cells: Mechanisms, Interactions and Biochemical Processes. What Did We Learn from Experimental Models? Pathogens. 2021 Feb 19;10(2):239. doi: 10.3390/pathogens10020239.)
A: Thank you for suggesting this manuscript, we analyzed it and included also as a reference as some data was also considered to be important also by us.
We hope that the revised form of the manuscript and our accompanying responses will be sufficient to make our manuscript suitable and accepted for publication in Journal of Clinical Medicine®. We shall look forward to hearing from you at your earliest convenience.
With our best regards,
Sincerely yours,
Rares Mircea Birlutiu, MD PhD
Victoria Birlutiu, Prof. Habil. MD. PhD
Reviewer 2 Report
I suggest a clarification about the excluded cultures: 7 cultures thought to be contaminated were excluded, along with other 6 positive cultures from the second stage of the 2-stage revision surgery? A clarification is needed about the 6 cases.
Author Response
Sibiu, 24.12.2022
To
the Editors of Microorganisms®
Dear Editor-in-Chief,
Dear Editor,
Dear reviewer,
Thank you for reviewing our manuscript. Please find attached a revised version of our manuscript, “Are there any changes in the causative microorganisms isolated in the last years from hip and knee periprosthetic joint infections?”.
Yours and the reviewers’ comments were highly insightful and enabled us to greatly improve the quality of our manuscript. We have modified the manuscript in response to the comments. Attached are our point-by-point response to each comment.
Reviewer 2 Comments:
I suggest a clarification about the excluded cultures: 7 cultures thought to be contaminated were excluded, along with other 6 positive cultures from the second stage of the 2-stage revision surgery? A clarification is needed about the 6 cases.
Answer: Thank you for taking from your precious time to be able to assess our manuscript. The comments were highly insightful and enabled us to improve our manuscript. Regarding the 6 cases that you mentioned, are 6 patients that underwent a 2SE revision strategy that during the second stage when a revision endoprosthesis was implanted (decision made based on the evolution of the case) had positive cultures from the intraoperative harvested specimens. We hope that we were able to the concern.
We hope that the revised form of the manuscript and our accompanying responses will be sufficient to make our manuscript suitable and accepted for publication in Microorganisms®. We shall look forward to hearing from you at your earliest convenience.
With our best regards,
Sincerely yours,
Rares Mircea Birlutiu, MD PhD
Victoria Birlutiu, Prof. Habil. MD. PhD